# A Snapshot of COVID-19 Incidence, Hospitalizations, and Mortality from Indirect Survey Data in China in January 2023 (Extended Abstract)

Juan Marcos Ramírez, Sergio Díaz-Aranda, Jose Aguilar, Antonio Fernández Anta

IMDEA Networks Institute, Madrid, Spain

Oluwasegun Ojo, Rosa Elvira Lillo

Universidad Carlos III, Madrid, Spain

## ABSTRACT

The estimation of incidence has been a crucial component for monitoring COVID-19 dissemination. This has become challenging when official data are unavailable or insufficiently reliable. Hence, the implementation of efficient, inexpensive, and secure techniques that capture information about epidemic indicators is required. This study aims to provide a snapshot of COVID-19 incidence, hospitalizations, and mortality in different countries in January 2023. To this end, we collected data on the number of cases, deaths, vaccinations, and hospitalizations among the fifteen closest contacts to survey respondents. More precisely, indirect surveys were conducted for 100 respondents from Australia on 19 January 2023, 200 respondents from the UK on 19 January 2023, and 1,000 respondents from China between 18-26 January 2023. To assess the incidence of COVID-19, we used a modified version Network Scale-up Method (NSUM) that fixes the number of people in the contact network (reach). We have compared our estimates with official data from Australia and the UK in order to validate our approach. In the case of the vaccination rate, our approach estimates a very close value to the official data, and in the case of hospitalizations and deaths, the official results are within the confidence interval. Regarding the remaining variables, our approach overestimates the values obtained by the Our World in Data (OWID) platform but is close to the values provided by the Officer of National Statistics (ONS) in the case of the UK (within the confidence interval). In addition, Cronbach's alpha gives values that allow us to conclude that the reliability of the estimates in relation to the consistency of the answers is excellent for the UK and good for Australia. Following the same methodology, we have estimated the same metrics for different Chinese cities and provinces. It is worth noting that this approach allows quick estimates to be made with a reduced number of surveys to achieve a wide population coverage, preserving the privacy of the participants.

## KEYWORDS

COVID-19, incidence estimation, indirect surveys, NSUM

## 1 INTRODUCTION

To effectively manage public health resources, monitoring infectious diseases such as COVID-19 requires knowledge of various epidemic indicators, such as the number of cases, deaths, and hospitalizations, among others. Most of these indicators have been collected through the use of methods that require the presence of a substantial portion of the target population, such as antigen test screenings or hospital records. In order to overcome these disadvantages, several methods have used direct surveys to estimate indicators [1, 2]. Unfortunately, direct surveys depend on the participation of a large number of people to obtain reliable estimates, usually collect sensitive personal data (which may deter respondents due to privacy concerns), and require careful data manipulation.

An alternative to these surveys is using indirect surveys, which ask participants about the people in their contact network, rather than themselves. From the responses provided by indirect surveys, the estimates of different variables can be derived using *Network Scale-up Method* (NSUM) [3, 4]. As a result of this approach, 1) a larger sub-population may be reached, 2) data collection costs may be reduced, 3) a computationally efficient method can be used to obtain estimates, and 4) participants will be assured of high levels of privacy. Indirect surveys have already been implemented for estimating indicators during the COVID-19 pandemic [5, 6].

In this work, we use indirect online surveys to capture a snapshot of cases, mortality, vaccination, and hospitalizations due to COVID-19 in China for the period of January 18-26, 2023. To this end, a modified version of the NSUM approach that fixes the number of people in the contact network is used to estimate different epidemic indicators. In essence, this modified version extracts knowledge about epidemic indicators without resorting to additional control questions that usually are considered to estimate the *reach* (the number of people in the contact network). In addition, a data pre-processing stage is included, which comprises of a set consistency filters and a nonlinear outlier detection stage, to improve the reliability of the collected data. We validate our approach using data from Australia and the United Kingdom (UK) collected on January 19, 2023. These metrics are compared with respect to the official values reported by Our World in Data (OWID) and the Office for National Statistics (ONS) from UK. In addition, we use Cronbach's alpha index [7], which is a reliability value to measure the internal consistency of the questionnaire generated by indirect surveys.

## 2 METHODS

### 2.1 Sampling Participants

We conducted online indirect surveys using the PollFish platform. Specifically, we conducted an online survey in China between January 18-26, 2023. This online survey collected information about various COVID-19 indicators (vaccination, deaths, and number of cases in the last month, the last 7 days, and the past 24 hours) among the 15 closest contacts of 1,000 participants (see Supplementary Information section for the English version of the survey questions). Notice that the selected number of closest contacts to respondents (15) is considered the size of the good-friends support group according to Dunbar's theory [8]. This number provides us a trade-off between the size of the subpopulation we aim to cover (reach) and

the minimization of undesired effects due to respondents such as transmission and recall errors [4]. Additionally, for validation, we conducted online surveys in Australia (100 responses) and the UK (200 responses) on January 19, 2023. Table 3 in Supplementary Information shows the characteristics of the survey respondents (the platform provides information on gender, age group, education, and ethnicity). The respondents of each survey are also stratified by region. For instance, Fig. 1 in Supplementary Information shows a map of China where the intensity corresponds to the number of questionnaires completed in each province.

## 2.2 Data Analysis

In order to obtain a reliable dataset, we performed two subphases of preprocessing: (1) an inconsistency filter, and (2) a univariate outlier detection.

(1) The inconsistency filter removes participants with inconsistent responses: less infected contacts than fatalities, less infected contacts than hospitalized, less infected contacts in the last month than in the last 7 days, and less infected contacts in the last month than in the last 24 hours.

(2) Since the collected variables exhibit extremely skewed distributions, the robust outlier detection method reported in [9] is applied. Based on the variable data, this method firstly estimates the quartiles $Q_1$ and $Q_3$, as well as the interquartile range ($IQR$). Then, the whiskers $Q_\alpha$ and $Q_\beta$ are set. Finally, this method preserves the samples in the interval limited by

$$[Q_1 - 1.5e^{aMC}IQR; \ Q_3 + 1.5e^{bMC}IQR] \tag{1}$$

where $MC$ is the medcouple statistic that estimates the degree of skewness of the data. Samples outside the interval are marked as outliers and, consequently, are removed. In addition, to estimate the parameters $a$ and $b$, we consider the system [9]

$$\begin{cases} \log\left(\frac{2}{3}\frac{Q_1-Q_\alpha}{IQR}\right) \approx aMC \\ \log\left(\frac{2}{3}\frac{Q_\beta-Q_3}{IQR}\right) \approx bMC. \end{cases} \tag{2}$$

where $Q_\alpha$ and $Q_\beta$ are the $\alpha$-th and $\beta$-th quantiles of the distribution, with $\alpha = 0.15$ and $\alpha = 0.85$.

We consider the NSUM approach to estimate the rates of the different COVID-19 indicators. In particular, NSUM is a statistical framework for estimating hidden populations from indirect surveys. There are three main NSUM approaches: frequentist models that estimate subpopulation rates, Bayesian models that include priors, and network models that estimate population properties [4]. To estimate cumulative incidences, hospitalization rates, and mortality rates, we modify an NSUM method belonging to the category of frequentist models based on the maximum likelihood estimation (MLE). In this regard, let $c_i$ be the number of contacts of the $i$-th respondent that have a particular characteristic, e.g., persons who have been hospitalized. Further, consider $r_i$ the number of close contacts of the $i$-th respondent (which in this study is fixed at $r_i = 15$, as shown in the questions in the Supplementary Information). The requirement of close contacts is introduced to minimize the effect of the visibility bias [10] with respect to the classical method [3]. Hence, we estimate the aggregated rate, $p$, as $\sum_i c_i / \sum_i r_i = \sum_i c_i/(15n)$, with $n$ as the number of responses (samples). The

estimator's variance is $\sqrt{p(1-p)/(15n)}$, assuming that the $c_i$ are independent binomial random variables with 15 trials and success probability $p$.

We evaluated the validity of our approach by comparing the difference between the official values reported on the *Our World in Data* (OWID)[1] platform and the values estimated by our approach for Australia and the United Kingdom (see Table 1). In both countries, official data were extracted between December 20, 2022, and January 19, 2023. In order to determine the number of hospitalized persons given the hospital occupancy, the length of a hospital stay is fixed at 4 days [12, 13].

Additionally, for the UK, we use the data provided by the *Office for National Statistics* (ONS)[2]. In particular, for the number of cases we use the daily estimates of the infected population obtained by the Coronavirus (COVID-19) Infection Survey of the ONS. For the 7 days and the last month's estimates, in order not to count multiple times the same cases, the sum of the daily percentages is divided by 10 days, an estimated average duration of the infection with Omicron [14]. Hospitalizations are the sum of the weekly admission rates with COVID-19 in England from Dec 19, 2022, to Jan 22, 2023 (5 weeks). Mortality is the rate of registered deaths involving COVID-19 in England from Dec 17, 2022, to Jan 20, 2023.

Finally, we use Cronbach's Alpha coefficient to measure the reliability of the results obtained from the indirect surveys. Specifically, it quantifies the reliability of a value of an unobservable variable constructed from the observed variables. The closer this coefficient is to its maximum value of 1, the greater the reliability of the measure, but in general, it is considered that values greater than 0.7 are sufficient to guarantee reliability. In this work, we compute Cronbach's Alpha coefficient based on correlations [15].

## 3 RESULTS

Table 1 displays the estimates and the 95% confidence interval for the surveys conducted in the UK and Australia. In addition, it shows the statistics provided by official reports. The confidence interval is computed as $p \pm 1.96\sqrt{p(1-p)/(15n)}$. As can be observed, the vaccination estimates are very close to the official values: they are estimated as 76.50% (73.70% - 79.29%) and 78.86% (95% confidence interval: 77.00% - 80.72%) in Australia and UK, respectively, while the official (OWID) values are 84.95% and 79.71%. In the case of mortality and hospitalizations in the last month, the official values are within the confidence interval of our estimates in the case of Australia. Specifically, the mortality rate is 0.34% (0.00% - 0.72%) and the official is 0.005%, the hospitalization rate is 1.02% (0.36% - 1.68%) and the official is 0.112%. Also, in the case of the UK, the official values of ONS are within the confidence interval of our estimates of the number of cases, new cases in the last 7 days, and cases in the last 24 hours. Cronbach's alpha coefficient is 0.83 for Australia and 0.95 for the UK, which tells us that the reliability of the estimates is very good. The results of the estimates and Cronbach's alpha coefficient allow concluding that we can use the indirect survey approach to make estimates when official data is not available or

---

[1]https://ourworldindata.org/, downloaded on July 24th, 2023. Observe that these values have changed from those downloaded in February 2023 [11].
[2]https://www.ons.gov.uk/, downloaded on February 3rd, 2023.

**Table 1: COVID-19 metrics in % (and 95% CI) obtained from indirect survey data and official reports for Australia and the UK. (1) People aged 12 years and over that have received at least one/two/three doses on Aug 31, 2022. (2) England data only, 5 weeks.**

| | Australia | | United Kingdom | | |
| --- | --- | --- | --- | --- | --- |
| | Indirect Survey | OWID | Indirect Survey | OWID | ONS |
| Cases (last month) | 12.43 (10.26 - 14.60) | 1.731 | 8.67 (7.39 - 9.96) | 0.298 | 9.663 |
| Vaccination rate | 76.50 (73.70 - 79.29) | 84.95 | 78.86 (77.00 - 80.72) | 79.71 | 93.6/88.2/70.2$^{(1)}$ |
| Mortality (last month) | 0.34 (0.00 - 0.72) | 0.005 | 0.43 (0.13 - 0.73) | 0.006 | 0.005$^{(2)}$ |
| Hospitalizations (last month) | 1.02 (0.36 - 1.68) | 0.112 | 0.81 (0.40 - 1.22) | 0.133 | 0.044$^{(2)}$ |
| Cases (24 hours) | 2.03 (1.10 - 2.96) | 0.118 | 1.30 (0.78 - 1.82) | 0.037 | 1.458 |
| New cases (7 days) | 2.71 (1.64 - 3.78) | 0.118 | 1.30 (0.78 - 1.82) | 0.023 | 1.116 |
| Cronbach's alpha | 0.83 | | 0.95 | | |

**Table 2: COVID-19 incidence metrics in % obtained from indirect survey data for China.**

| | | Samples | Cases (last month) | Vaccination rate | Mortality (last month) | Hosp (last month) | Cases (24 hours) | Cases (7 days) |
| --- | --- | --- | --- | --- | --- | --- | --- | --- |
| **China** | | 469 | **78.57 (77.62-79.54)** | **91.03 (90.36-91.70)** | **1.19 (0.94-1.45)** | **9.30 (8.61-9.97)** | **2.87 (2.48-3.26)** | **9.52 (8.83-10.21)** |
| Provinces | Jiangsu | 48 | 75.56 (72.42-78.69) | 87.92 (85.54-90.30) | 1.67 (0.73 - 2.60) | 7.64 (5.70-9.58) | 2.64 (1.47-3.81) | 9.44 (7.31-11.58) |
| | Guangdong | 45 | 80.00 (76.98-83.02) | 86.07 (83.46-88.69) | 0.59 (0.01-1.17) | 5.33 (3.64-7.03) | 3.26 (1.92-4.60) | 6.96 (5.04-8.88) |
| | Shandong | 27 | 74.81 (70.59 - 79.04) | 95.80 (93.85-97.76) | 1.48 (0.30-2.66) | 8.40 (5.69-11.10) | 2.22 (0.79-3.66) | 6.67 (4.24-9.10) |
| Cities | Shanghai | 9 | 68.89 (61.08-76.70) | 88.15 (82.70-93.60) | 2.22 (0.00-4.71) | 5.93 (1.94-9.91) | 0.74 (0.00-2.19) | 5.19 (1.44-8.93) |
| | Guangzhou | 11 | 81.82 (75.93-87.70) | 86.67 (81.48-91.85) | 1.82 (0.00-3.86) | 9.70 (5.18-14.21) | 4.85 (1.57-8.13) | 7.27 (3.31-11.24) |
| | Chengdu | 8 | 89.17 (83.61-94.73) | 88.33 (82.59-94.08) | 0.83 (0.00-2.46) | 8.33 (3.39-13.28) | 0.83 (0.79-2.45) | 8.33 (3.39-13.28) |
| | Beijing | 8 | 74.17 (66.33-82.00) | 91.67 (86.72-96.61) | 0.83 (0.00-2.45) | 13.33 (7.25-19.42) | 5.00 (1.10-8.90) | 11.67 (5.92-17.41) |

reliable and use them considering a prudential bias when assessing them.

Table 2 shows the estimated results for China for all the questions of the survey. While 1.000 indirect survey responses were collected, the filters specified in Section 2.2 were used, reducing drastically the sample size to 469. Comparing our results with the OWID data for China, the vaccination rate is 91.9% while we estimate 91.03% (90.36%-91.7%), which is almost a perfect match. The number of deaths reported by OWID is 0.005% while we estimate 1.19% (0.94%-1.45%), a much higher value. However, OWID warns that "the number of confirmed deaths may not accurately represent the true number of deaths". Therefore, our estimate could serve as a first approximation (that may be biased). Our estimate of the number of cases in the last month is 78.57% (77.62%-79.54%), very far from 6.182% reported by OWID (which warns that "the number of confirmed cases is lower than the true number of infections"). Note that some areas of China may have a high incidence, as noted in the report published at [16]: "nearly 90% of Henan's population had been infected by 6 January".

We compute estimates for the provinces and cities with the largest number of samples (see Table 2). The rate of vaccination and cases in the last month is similar in all of them and similar to the values in China. The Guangdong province shows the lowest estimates of hospitalizations and deaths, while it has large case estimates

among provinces. Among cities, Beijing shows low estimates of monthly cases, but large rates of recent cases and hospitalizations. Unfortunately, the sample size for cities is very small. Finally, we would like to point out that, in general, the data are relatively small compared to the size of the country. Additionally, as can be seen in Table 3 in Supplementary Information, the sample is biased by age and education level. These biases are reduced with the use of indirect questions, but still more studies are needed.

## 4 CONCLUSIONS AND FUTURE WORK

This work aims to estimate a snapshot of COVID-19 incidence, hospitalizations, and mortality from indirect surveys in China in January 2023. To estimate these epidemic indicators, we used a modified version of the NSUM technique that fixes the number of people in the contact network. In addition, a data pre-processing stage is included to extract a reliable set of survey samples. In future work, we are interested in analyzing multiple data preprocessing techniques to minimize the number of discarded samples and maximize indirect survey knowledge extraction. Additional results and a more extended discussion can be found in the full version of the article [11].

## 5 RESEARCH ETHICS APPROVAL

To carry out this, a request was previously made before the ethics committee of IMDEA Network Institute, who approved it in the

last quarter of 2022. Basically, the ethics committee approved that the study could be carried out keeping the anonymity of the respondents. On the other hand, the platform used for the collection of survey information guarantees that the participants (belong to that platform) give their consent to participate in them.

## 6 CONFLICT OF INTEREST DISCLOSURES

None reported.

## 7 FUNDING/SUPPORT

This work was partially supported by grants COMODIN-CM and PredCov-CM, funded by Comunidad de Madrid and the European Union through the European Regional Development Fund (ERDF), and grants TED2021-131264B-I00 (SocialProbing) and PID2019-104901RB-I00, funded by Ministry of Science and Innovation - State Research Agency, Spain MCIN/AEI/10.13039/ 501100011033 and the European Union "NextGenerationEU"/PRTR.

## 8 DATA SHARING STATEMENT:

The data collected in the indirect surveys is publicly available at https://github.com/GCGImdea/coronasurveys/tree/master/papers/2023-COVID-19-China-January.

## 9 ACKNOWLEDGMENT:

We want to thank Lin Wang for his help with the Chinese version of the survey.

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

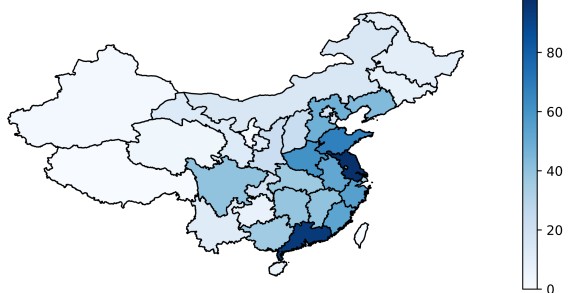

**Figure 1: Number of completed questionnaires for the survey deployed in China**

**Table 3: Characteristics of the survey respondents for Australia, the United Kingdom, and China.**

| | Characteristic | Australia | United Kingdom | China |
|---|---|---|---|---|
| 1. | Number of participants | 100 | 200 | 1000 |
| 2. | Gender, (%) | | | |
| | (a) Female | 56.00 | 58.00 | 46.90 |
| | (b) Male | 44.00 | 42.00 | 53.10 |
| 3. | Age groups, (%) | | | |
| | (a) 18-24 | 13.00 | 9.50 | 18.70 |
| | (b) 25-34 | 27.00 | 26.00 | 44.30 |
| | (c) 35-44 | 29.00 | 24.50 | 27.40 |
| | (d) 45-54 | 17.00 | 22.50 | 8.40 |
| | (e) >54 | 14.00 | 17.50 | 1.20 |
| 4. | Education, (%) | | | |
| | (a) Middle school | 2.00 | 5.00 | 1.50 |
| | (b) High school | 33.00 | 22.00 | 7.90 |
| | (c) Technical college | 14.00 | 35.00 | 8.30 |
| | (d) University | 43.00 | 25.00 | 63.30 |
| | (e) Post-graduate | 7.00 | 11.50 | 18.90 |
| 5. | Ethnicity, (%) | | | |
| | (a) Arab | 0.00 | 0.00 | 0.20 |
| | (b) Asian | 8.00 | 7.50 | 94.60 |
| | (c) Black | 0.00 | 2.50 | 0.20 |
| | (d) Hispanic | 0.00 | 1.00 | 0.00 |
| | (e) Latino | 0.00 | 0.00 | 0.20 |
| | (f) White | 83.00 | 74.00 | 1.00 |
| | (g) Multiracial | 3.00 | 1.00 | 0.20 |
| | (h) Other | 6.00 | 14.00 | 1.60 |

## SUPPLEMENTARY INFORMATION
## Questions of the Indirect Survey

*Questions in English.* Think of your 15 closest contacts in the last month. The rest of the questions below are with respect to this group of people. These contacts can be family, friends, or colleagues whose health status you know.

(1) From the above 15 closest contacts in the last month, how many have had COVID-19 in the last month?

(2) From the above 15 closest contacts in the last month, how many have been hospitalized for COVID-19 in the last month?

(3) From the above 15 closest contacts in the last month, how many died from COVID-19 in the last month?

(4) From the above 15 closest contacts in the last month, how many have COVID-19 today?

(5) From the above 15 closest contacts in the last month, how many started with COVID-19 in the latest 7 days?

(6) From the above 15 closest contacts in the last month, how many have (ever) been vaccinated for COVID-19?