# OpenReview forum: "A Snapshot of COVID-19 Incidence, Hospitalizations, and Mortality from Indirect Survey Data in China in January 2023 (Extended Abstract)"
_KDD.org/2023/Workshop/epiDAMIK — KDD 2023 Workshop epiDAMIK_

### Official Review · Reviewer_kBH4 · 2023-06-20
**The paper proposes an indirect survey method and the Network Scale-up Method (NSUM) to estimate COVID-19 indicators. While the methodology and data preprocessing are explained well, the lack of detailed discussion on NSUM and omission of related works utilizing NSUM raise concerns about the novelty and completeness of the research.**

**Rating:** 2
**Confidence:** 4

**Review:**

In this paper's introduction, the challenge of obtaining reliable COVID-19 data is explained due to the need for target data and costly tests. To address this issue, the authors suggest utilizing an indirect survey method that involves a fixed number of participants. They also propose the NSUM method to estimate various epidemic indicators. However, I have noticed that the NSUM method for constructing a contact network in COVID-19 cases has been utilized before but is not mentioned in the related works.

The methodology used in the paper is reliable. The survey gathered data from 15 contacts of 1000 participants concerning various COVID-19 indicators. However, the paper did not specify how the participants were selected or if their contacts were mutually exclusive. For instance, if the participants were hospital staff, their contacts would differ from those who work remotely. The paper clearly explained the data preprocessing, utilizing inconsistency filters and univariate outlier detection to eliminate anomalies while accounting for skewed data. Nevertheless, the NSUM method was not extensively discussed.

In the methodology section, the researchers assessed their findings in the UK and Australia by contrasting them with the official results on the OWID platform. Although the mortality rate differed significantly, Cronbach’s alpha score was high, indicating strong internal consistency. Notably, the vaccination rate result closely matched the actual value. In my opinion, the sample size should have been larger as the filters have significantly decreased the number of sample surveys available.

The paper is well-organized, with each section thoroughly explained. However, I noticed that the methodology of NSUM is missing, which plays a vital role in constructing the network graph and producing the results. I also came across some related works that utilize NSUM from survey data in COVID-19 cases, which were not included in the relevant work section. Although the problem they were addressing is undoubtedly significant, I remain somewhat skeptical about the novelty of this work.

This paper does not heavily rely on mathematical formulas or algorithms, making it less technical in nature. However, the equations presented in the paper seem to be accurate to my understanding.

To summarize, the paper is well-written and presents a clear problem definition. However, the methodology could use further development, and there is a lack of reference to recent work on the same problem.

---

### Official Review · Reviewer_K1rx · 2023-06-29
**Review of the paper.**

**Rating:** 4
**Confidence:** 5

**Review:**

This study presents estimates for COVID incidence cases, deaths, and vaccination rates based on a survey study.

Overall, the paper is really good in quality, clarity, originality and significance. The paper is well-written; however, there are a few areas that the authors should address:

1. In section 2.1, the authors conducted an online survey in Australia and the UK for validation. It would be beneficial for the authors to provide justification for selecting these specific countries. For example, they should explain why China was not included in the online survey.

2. If space allows, it would be helpful to include a figure illustrating the skewness in the data, as discussed in Section 2.2. This figure could demonstrate the requirement of Medcouple statistics.

3. Please include a reference for the Cronbach's alpha coefficient. This would provide readers with additional information and support the use of this measure.

4. To  enhance the paper's transparency, the authors should clarify how the 95% confidence interval (C.I.) was computed in Table 1 and Table 2.

5. It is unclear how a small sample size, such as the one used for all the cities, can be utilized to derive the confidence interval and make any valid claims.

By addressing these points, the authors can further improve the clarity and comprehensiveness of their work.

---

### Official Review · Reviewer_H35u · 2023-06-29
**Review of a Paper on Estimating COVID-19 Snapshots: Strong Results, Need for Comparisons, and Requirement for Further Elaboration**

**Rating:** 2
**Confidence:** 2

**Review:**

Quality:

The quality of the paper is good overall. The authors present an approach to estimating COVID-19 snapshots using a modified Network Scale-up Method (NSUM) and validate their estimates against official data. The data preprocessing stage helps enhance the reliability of the collected data, and the privacy preservation aspect adds value to the study. However, there are some limitations, such as the lack of comparisons with other estimation methods or its limited generalizability.

Clarity:

The paper is generally well-written and presents the information in a clear manner but does have a few typos and items which could have been explained some more. The introduction provides adequate background information about the need for indirect survey methods and the challenges associated with official COVID-19 data. The methodology section explains the data preprocessing techniques well, but could benefit from further clarification. For example, the NSUM technique is only cited but not explained anywhere, and also the choice of setting ri=15 is not justified (why not 5 or 10?).

Originality:

The paper cites that the use of indirect surveys to estimate different variables using NSUM is not something new, it also cites that this has also been done for estimating different indicators during the COVID-19 pandemic.

Significance:

The significance of this work lies in its potential to provide valuable insights into COVID-19 indicators, especially in settings where official data is limited or unreliable. The indirect survey method offers a practical solution to estimate important epidemiological information, which can aid decision makers and researchers in understanding the spread of the disease and acting accordingly. The paper's comparison with official data and validation of estimates add credibility to its findings, further highlighting its significance.

Pros:

- Justification for Indirect Surveys: The paper provides a strong rationale for using indirect surveys, highlighting privacy preservation and other benefits.

- Validated and Discussed Results: The paper presents well-validated results and provides a comprehensive discussion of the findings.
Use of Cronbach's Alpha Coefficient: The paper employs Cronbach's alpha coefficient, a reliable measure of internal consistency, enhancing the robustness of the analysis.

- Acknowledgment of Sample Size Limitation: The paper recognizes the limitation of the sample size and discusses its potential impact on the accuracy and generalizability of the estimates.

- Data Preprocessing Stage: The paper includes a well-described data preprocessing stage, which enhances the reliability and quality of the collected data.


Cons:

- Need for Comparison to Validate Modifications and NSUM Choice: The paper should include a comparison with other methods to validate the modifications made and the selection of the Network Scale-up Method (NSUM).

- Insufficient Elaboration on NSUM and Choice of "ri": The paper should provide more explanation and elaboration on NSUM and the selection of "ri" to improve reader understanding.

- Limited Generalizability: The study's focus on a specific time period and a restricted set of countries (China, Australia, and the UK) limits the generalizability of the results to other countries and different time periods.

- Few typos: 1- Typo in mortality rate, should be 0.72 not 0.22 based on table (line 218 right column).  2- Variable naming is either not consistent or not explained sufficiently in equations 1 and 2, would be good to clarify here what the "a", "b", "alpha", and "beta" variables represent


In summary, this paper presents a good approach to estimate COVID-19 indicators using the Network Scale-up Method. While it has strong results, there are also limitations to consider. Further elaboration could address a lot of these limitations.

---

### Official Review · Reviewer_RNUi · 2023-06-30
**Paper provides a succinct and accessible method to estimate disease incidence**

**Rating:** 4
**Confidence:** 4

**Review:**

### Summary
This paper seeks to improve disease incidence estimation methods using information from surveys about contacts, rather than the respondents direct experience. They can obtain much more information by asking about multiple individuals the respondent knows about rather than gather information about only one individual per survey. From this information, they use a modified network scale up method to determine estimated incidence for Australia, the UK, and China and use Cronbach’s alpha to verify the reliability of their data. In addition, they thoroughly clean the data they obtain in order to get a better estimate.

### Strengths
- They compare with a range of locations for validation rather than relying on only one.
- Their data-preprocessing and estimation methods are clear and well-explained.

### Weaknesses
- The authors do not discuss how the differences in region of respondents affects the estimates in other regions. For example, how do estimates based on the regions with many respondents perform for regions with a much lower response rate?
- They do not discuss the impact of sample size. Can a study be performed where the sample size is discussed in the context of confidence and estimate performance? It may not be viable to study, but are there hypotheses on when the sample size is too large (i.e., the sets of 15 contacts begin to overlap resulting in over-counting)?

### Suggestions
- The sentence “…and hospitalizations among 15 of the closest contacts” could use a bit more elaboration, such as “…closest contacts to survey respondents”.
- What are the results if the data was not pre-processed?

### Minor
- What is n on line 177?
- The writing is imprecise in some places such as line 206, 224

---

### Official Review · Reviewer_TRCH · 2023-06-30
**Paper shows the efficacy of indirect surveys in the estimation of epidemic indicators in places where official figures are unreliable.**

**Rating:** 4
**Confidence:** 3

**Review:**

This paper tackles the problem of estimating snapshot Covid-19 incidence rates in locations where the official figures are believed to be unreliable. They utilize an indirect survey method to collect data from respondents which has the benefits of preserving their privacy and mitigating bias due to age or education level. They modify the Network Scale Up Method by fixing the number of close contacts in their survey. They validate their approach by estimating for the UK and Australia using the English version of the indirect survey and present results from China.

I think this is well-written paper describing the methods, data collection strategy and prior related work in adequate detail. By comparing their estimates for the UK and Australia with the official figures, they show the validity of their estimates in China where the official figures might conceal the true rates of hospitalizations and mortality. The results are very interesting as they show a general agreement with the official vaccination rates while showing wide disparity in the estimates for deaths and cases.

The data pre-processing steps weeds out inconsistent and/or outlier responses. This whittles down the sample size from 1000 to 469. This affects the ability to reliably estimate for cities, especially considering the population size. I was wondering if there was a way to preserve some of the inconsistent responses by making expert adjustments and how that would affect the results?

Lastly, they compute the Cronbach's Alpha coefficient on the responses of the indirect surveys for the UK and Australia, which suggests that the indirect survey method is reliable. I believe the methods in this paper are well thought-out and the results are worth a close look. I await the outcome of their future work.